# Interleukin-36 cytokines alter the intestinal microbiome and can protect against obesity and metabolic dysfunction

Eirini Giannoudaki[1,2,6], Yasmina E. Hernandez-Santana [1,2,6], Kelly Mulfaul[1,2], Sarah L. Doyle[1,2], Emily Hams[1], Padraic G. Fallon [1,2], Arimin Mat[3], Donal O'Shea[3], Manfred Kopf[4], Andrew E. Hogan[2,5] & Patrick T. Walsh [1,2]

Members of the interleukin-1 (IL-1) family are important mediators of obesity and metabolic disease and have been described to often play opposing roles. Here we report that the interleukin-36 (IL-36) subfamily can play a protective role against the development of disease. Elevated IL-36 cytokine expression is found in the serum of obese patients and negatively correlates with blood glucose levels among those presenting with type 2 diabetes. Mice lacking IL-36Ra, an IL-36 family signalling antagonist, develop less diet-induced weight gain, hyperglycemia and insulin resistance. These protective effects correlate with increased abundance of the metabolically protective bacteria *Akkermansia muciniphila* in the intestinal microbiome. IL-36 cytokines promote its outgrowth as well as increased colonic mucus secretion. These findings identify a protective role for IL-36 cytokines in obesity and metabolic disease, adding to the current understanding of the role the broader IL-1 family plays in regulating disease pathogenesis.

[1] Department of Clinical Medicine, School of Medicine, Trinity Translational Medicine Institute, Trinity College Dublin, Dublin 8, Ireland. [2] National Children's Research Centre, Our Lady's Children's Hospital, Crumlin, Dublin 12, Ireland. [3] Obesity Immunology Research, St Vincent's University Hospital and University College Dublin, Dublin 4, Ireland. [4] Department of Biology, Molecular Biomedicine, Institute of Molecular Health Sciences, ETH Zurich, Zurich, Switzerland. [5] Department of Biology, National University of Ireland, Maynooth, Ireland. [6] These authors contributed equally: Eirini Giannoudaki, Yasmina E. Hernandez-Santana. Correspondence and requests for materials should be addressed to P.T.W. (email: walshp10@tcd.ie)

Over the past 40 years, global levels of obesity have more than doubled. As obesity predisposes to the metabolic syndrome and has been definitively linked to coronary heart disease, stroke, type 2 diabetes and certain forms of cancer, this growing epidemic represents one of the most significant current global health challenges.

In tandem with the emergence of this problem has been an increase in understanding the pathological mechanisms which link the obese state to the development of disease. Central to these mechanisms is the heightened state of systemic inflammation as a result of obesity. Since the seminal observations made by Hotamisligil et al., demonstrating the role of adipose tissue-derived TNF as an inhibitor of insulin signalling and first linking obesity-driven inflammation to type 2 diabetes[1], it is now recognised that the nature of adipose tissue inflammation is a major contributor towards obesity-related diseases[2–4]. Further complexity is added through the instructive role of the gastrointestinal immune system in influencing the development of inflamed adipose tissue. It is now evident that changes in the homoeostatic barrier function of the gut can play a major role in the development of systemic inflammation in obesity[5]. In this context, it is perhaps unsurprising that the constituents of the intestinal microbiome are also emerging as an important factor in this regard. In healthy individuals, the gastrointestinal immune system plays an immuno-surveillant role, in constant interaction with the intestinal microbiome which profoundly influences its activity. The role of the microbiota in the development of obesity-driven metabolic disease first emerged from studies on germ-free mice, which demonstrated reduced levels of experimental weight gain and improved glucose tolerance[6,7]. Moreover, the transfer of an 'obese' microbiome to germ-free mice can lead to significant increases in adiposity confirming the important instructive role in contributing to disease[8]. Similarly, changes in the composition of the intestinal microbiome have been associated with obesity and the development of metabolic disease in humans[9,10]. As well as contributing to obesity and disease, it is also becoming apparent that certain constituents of the microbiota can act opposingly to restrict adiposity. In recent times, it has been demonstrated that the presence of the bacteria Akkermansia muciniphila in the intestinal microbiome can act directly to suppress weight gain and impaired glucose tolerance in mice[11]. Many of these protective effects have now been recapitulated using a purified outer membrane protein, Amuc_1100, derived from A. muciniphila[12]. It is also noteworthy that several dietary interventions targeting obesity and glucose intolerance, and the glucose-lowering drug metformin, increase A. muciniphila abundance[11,13–15]. Significantly, follow-up studies have also demonstrated a clear association between the relative abundance of this bacterium in the microbiome and the metabolic health of obese patients[16]. Such observations suggest that the composition of the intestinal microbiome can be altered towards a more 'healthy' lean state, highlighting its potential as a therapeutic strategy, and placing the identification of the mechanistic pathways which can alter the composition of the microbiota as a particular area of interest.

In this study, we identify the IL-36 family of cytokines as one such pathway. The IL-36 family of cytokines are a recently described subset of the larger IL-1 family which are emerging as important mediators of obesity-related metabolic disease[17]. The family consists of three separate agonistic ligands, designated IL-36α, IL-36β and IL-36γ, and a specific IL-36 receptor antagonist (IL-36Ra), all of which act through a specific IL-36 receptor[18]. Similar to the more extensively characterised 'classical' IL-1 cytokines, IL-1α and IL-1β, IL-36 cytokines are thought to act as important mediators of homoeostasis and inflammation, but in a more tissue-restricted manner. In this role, IL-36 cytokines are known to play a central role in orchestrating psoriatic inflammation in the skin and can also act as mediators of gastrointestinal inflammation and homoeostasis[19–23]. In contrast to these established roles of the IL-36 family, little information is available concerning how, and if, this cytokine family may influence obesity-induced systemic inflammation leading to metabolic syndrome. Related IL-1 family members have previously been studied in this context revealing opposing functions[17,24]. For example, IL-1β has long been implicated in the pathogenesis of both type 2 diabetes and atherosclerosis, and Canakinumab (anti-IL-1β-neutralising monoclonal antibody) is currently the focus of extensive clinical investigation for these indications[17]. In direct contrast, IL-18 and IL-33 have been demonstrated to play a protective role in animal models of obesity-driven metabolic syndrome, although the precise mechanisms through which this occurs have not been identified[17,24]. Moreover, there are currently no data to indicate that the microbiome may play a role in these protective effects.

In this study, we demonstrate that IL-36γ expression is increased in the serum of clinically obese patients and these elevated expression levels are negatively correlated with both haemoglobin A1c (HbA1c) and fasting blood glucose (FBG) levels among patients with type 2 diabetes indicating a protective role for these cytokines in countering metabolic dysfunction. These protective effects are mirrored in mice with a deficiency in the IL-36 receptor antagonist (Il36rn−/−), which exhibit reduced weight gain and metabolic dysfunction. Mechanistically, the protective effects occur in association with the relative outgrowth of the commensal bacterial strain A. muciniphila in the intestinal microbiome which is facilitated through IL-36-mediated enhanced mucus secretion in the colon.

## Results

**Elevated IL-36 cytokine expression among obese patients.** In an effort to determine whether the IL-36 family of cytokines may play a role in the development of obesity and metabolic disease, we investigated serum expression levels of the IL-36 family among a cohort of adult patients presenting with clinical obesity (BMI>30 kg/m²), with or without signs of diabetes, as determined by serum HbA1c levels, according to American Diabetes Association guidelines[25]. Expression of all IL-36 family ligands was detected among a small number of lean patients. Interestingly, levels of IL-36γ alone were found to be specifically and significantly elevated among obese patients when compared with healthy lean control subjects (Fig. 1c). These elevated levels were observed among obese patients irrespective of whether they exhibited evidence of type 2 diabetes (i.e. HbA1c ≥ 48 mmol/mol), indicating that elevated IL-36γ expression is directly associated with the obese phenotype. In contrast, no such differences were observed in the expression levels of the other agonistic ligands, IL-36α and β (Fig. 1a, b). Expression of the IL-36 receptor antagonist was low or not detected among all subjects examined (Fig. 1d).

To further determine what impact, if any, increased IL-36γ might play in the pathogenesis of obesity and metabolic disease, we examined whether elevated expression levels were correlated with established clinical and biochemical parameters of disease recorded for each patient subgroup. These parameters included sex, age, BMI, weight in kg, HbA1c and FBG levels, as well as serum lipid levels including cholesterol, triglycerides, LDL and HDL (Supplementary Table 1). Interestingly, this analysis revealed a strong negative correlation between blood glucose levels as determined by both HbA1c ($r = -0.74$, $p = 0.0015$) and FBG ($r = -0.54$, $p = 0.03$), and serum levels of IL-36γ, specifically among obese patients with type 2 diabetes (HbA1c ≥ 48 mmol/mol) (Fig. 1e, f). No such correlation was found to exist among obese patients that did not have diabetes, with HbA1c levels less than the 48 mmol/mol cut-off (Fig. 1g, h). These data demonstrate that elevated IL-36γ levels are associated

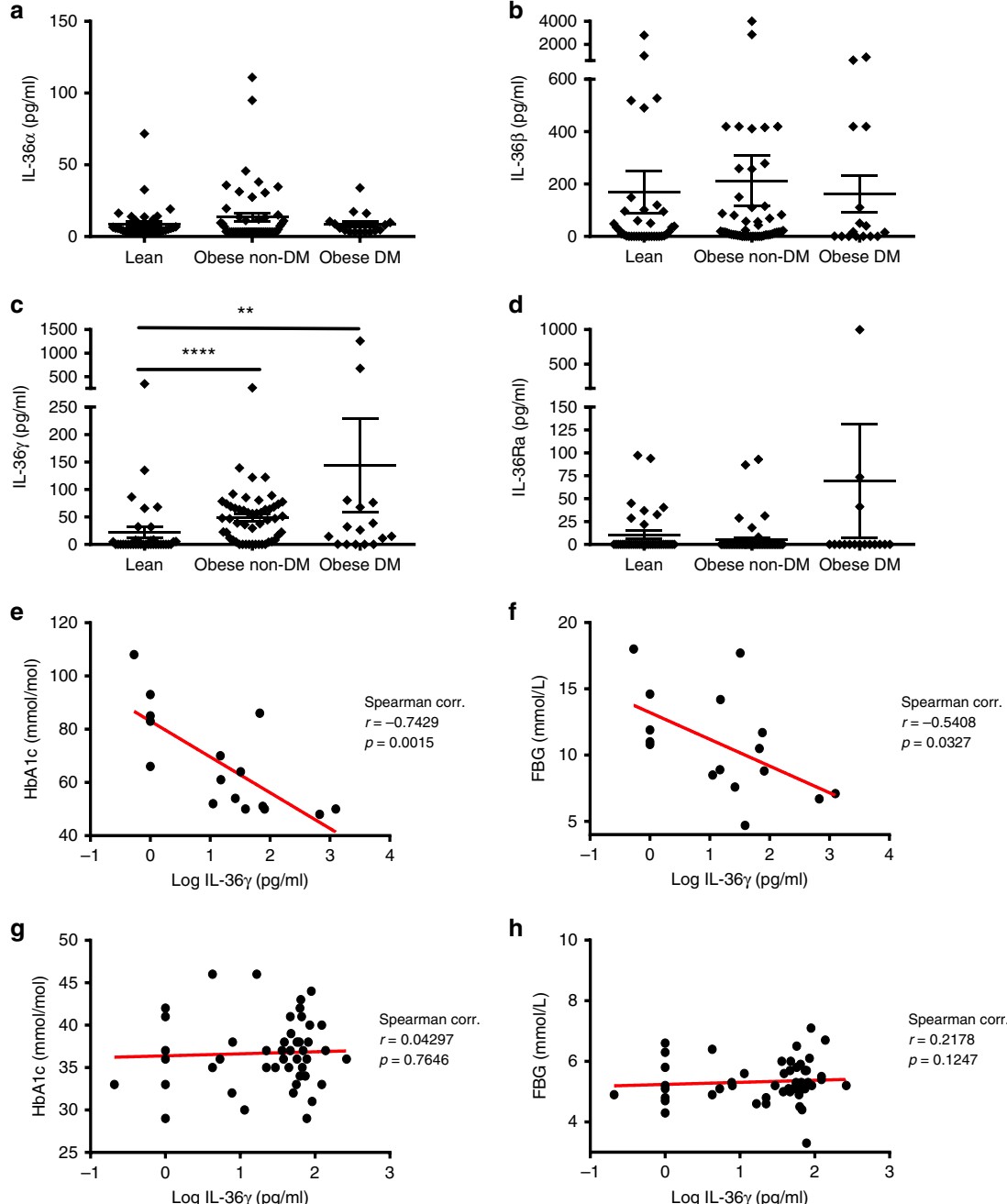

**Fig. 1** IL-36 cytokine expression levels in the serum of obese patients. **a–d** IL-36α, β, γ and IL-36Ra levels in the serum of lean individuals ($n = 37$), obese non-diabetic patients (HbA1c < 48) ($n = 51$) and obese with diabetes (HbA1c ≥ 48) ($n = 16$), as measured by ELISA. **e–h** Correlation analysis of IL-36γ levels with HbA1c and FBG in the serum of diabetic obese patients (**e**, **f**) and non-diabetic obese patients (**g**, **h**). Spearman correlation coefficient ($r$) and the corresponding $p$ value shown. For (**a–d**) data are shown as means ± SEM, statistical analysis by two-tailed Mann–Whitney test, **$p < 0.01$, ****$p < 0.0001$. DM: diabetes mellitus. Source data are provided as a Source Data file

with lower blood glucose levels among obese patients with diabetes and indicate that IL-36 cytokines may act to promote metabolic health in obesity.

**Reduced obesity and metabolic disease in *Il36rn*−/− mice.** To gain a greater understanding of how the IL-36 family of cytokines might impact the pathogenesis of obesity and metabolic disease, we examined mice deficient in the *Il36rn* gene which encodes the IL-36 receptor antagonist[19]. Interestingly, under normal chow-feeding conditions, these mice exhibited progressively less weight

gain compared with wild-type control mice (wt), which was evident by 6 months of age (Fig. 2a), despite similar levels of food consumption (Fig. 2b). This decreased weight gain was associated with a lower mass of both epidydimal (EAT) and subcutaneous (SAT) white adipose tissue depots (Fig. 2c). Furthermore, *Il36rn*−/− mice were found to have significantly improved glucose tolerance and lower insulin resistance at 10 months of age over their wt counterparts (Fig. 2d–g), demonstrating that loss of expression of the IL-36 receptor antagonist leads to a decrease in normal weight gain and the spontaneous development of glucose intolerance observed in aged mice. Moreover, younger *Il36rn*−/−

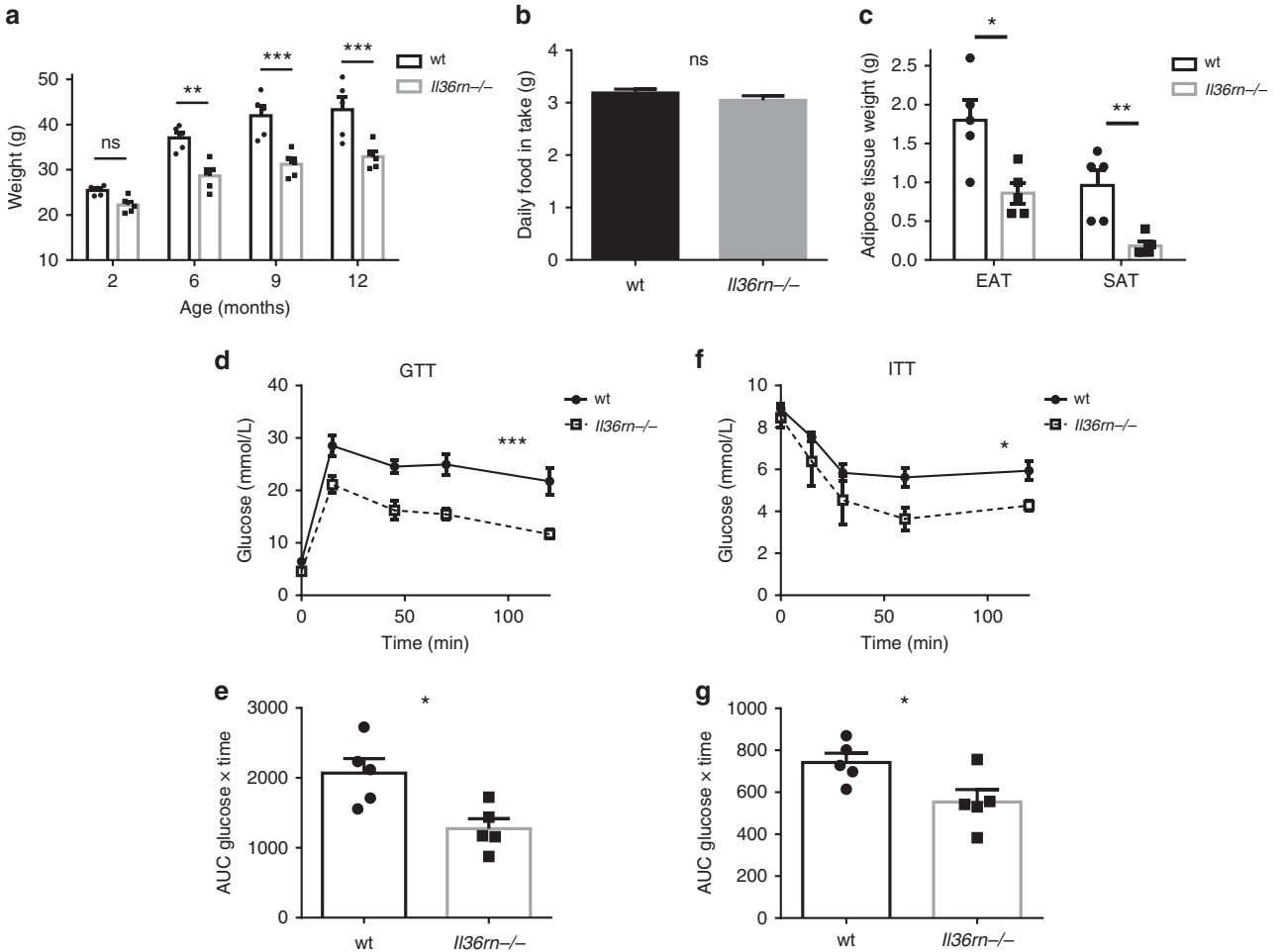

**Fig. 2** *Il36rn−/−* mice are protected from spontaneous weight gain and metabolic dysfunction. **a–e** Male *Il36rn−/−* and wt control mice of the same age were kept on normal chow diet up to 12 months of age ($n = 5$ per group). Weights (**a**) and daily food intake per mouse (**b**) were monitored over time from 2 months of age. **c** EAT and SAT depots mass was measured at 12 months of age. **d**, **e** Intraperitoneal glucose tolerance test (GTT) performed at 9 months of age; glucose measurements over time after glucose bolus-injected i.p. (**d**) and area under curve (AUC) (**e**) shown. **f**, **g** Intraperitoneal insulin tolerance test (ITT) performed at 9 months of age; glucose measurements over time after insulin bolus-injected i.p. (**f**) and AUC (**g**) shown. Data represent means ± SEM. Statistical analysis using two-tailed unpaired student's *t*-test, or RM two-way ANOVA with Bonferroni's correction for multiple comparisons for (**a**), (**d**) and (**f**). ns $p > 0.05$, *$p < 0.05$, **$p < 0.01$, ***$p < 0.001$. Source data are provided as a Source Data file

mice (2 months old), with comparable starting weights to their wild-type counterparts, exhibited significantly less weight gain and adipose tissue accumulation when placed on a high-fat diet (HFD) (60 kcal% fat) for 8–10 weeks (Fig. 3a, c). Again, this occurred in the absence of any alteration in daily food intake (Fig. 3b). These mice also displayed significant protection from HFD-induced metabolic dysfunction as determined by their fasting serum insulin levels, which were significantly lower in the *Il36rn−/−* mice (Fig. 3h), as well as glucose and insulin tolerance tests (Fig. 3d–g). In addition, *Il36rn−/−* mice exhibited increased Akt phosphorylation in the liver, after insulin challenge, indicating enhanced insulin sensitivity in this tissue compared with wt mice (Fig. 3i, j). Phosphorylation of the insulin receptor β (IR) also appeared to be slightly higher in the liver of the *Il36rn−/−* mice, albeit not to a significant extent (Fig. 3i, j). No differences in insulin sensitivity were detected in either muscle or EAT under these conditions (Fig. 3i, k, l). These data indicate that unrestricted IL-36 cytokine signalling, through deficiency of the IL-36 receptor antagonist, suppresses weight gain and the development of metabolic dysfunction in a diet-induced obesity mouse model.

In contrast, mice with a global deficiency in *Il1rl2*, the gene encoding the IL-36 receptor, exhibited normal weight gain and comparable levels of glucose and insulin tolerance to those

observed in wt mice after 8–10 weeks exposure to HFD (Supplementary Fig. 1). Together, these data suggest that while constitutive expression of the IL-36 receptor does not play a role in HFD-induced adiposity in mice, hyperactive IL-36 family activity can protect against obesity and metabolic disease.

**The intestinal microbiome is altered in *Il36rn−/−* mice.** As we observed that IL-36 receptor antagonist deficiency improves glucose tolerance in mice, we next examined whether adipose tissue inflammation induced upon exposure to HFD was altered in this setting (Supplementary Fig. 2). Although the absolute numbers of adipose tissue infiltrating CD45+ cells were reduced in *Il36rn−/−* mice after 10 weeks exposure to HFD (Supplementary Fig. 2a), as were the numbers of F4/80+ CD11b+ total macrophages (Supplementary Fig. 2b) and F4/80+ CD11b+ CD11c+ CD301− M1 macrophage subset (Supplementary Fig. 2c), these differences were not evident when analysed per gram of adipose tissue indicating that *Il36rn* deficiency did not quantitatively affect the degree of CD45+ cell infiltration observed (Supplementary Fig. 2e). In addition, no differences in the relative numbers of infiltrating pathogenic or homoeostatic macrophage subsets were observed (Supplementary Fig. 2f–h), indicating that

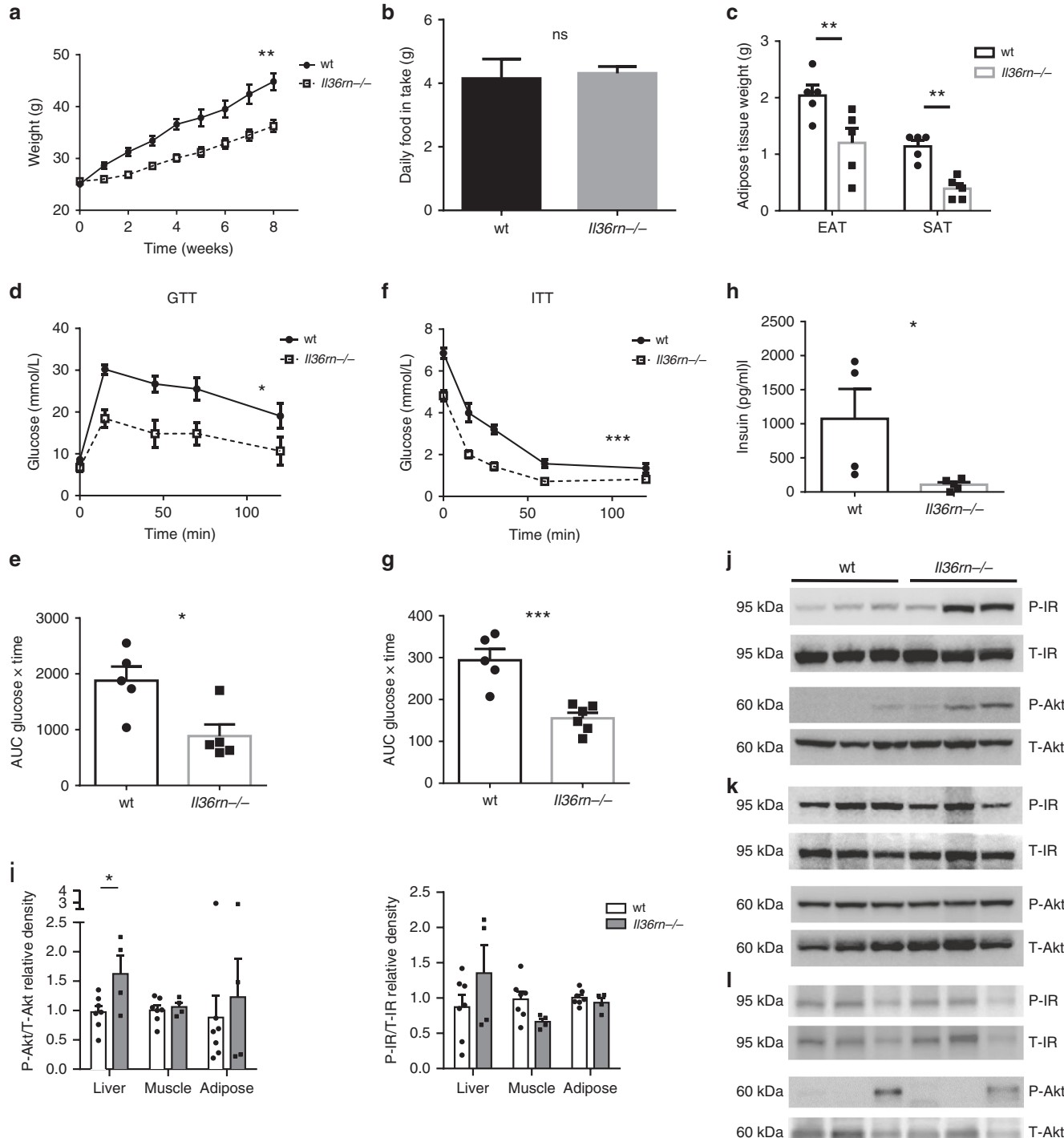

**Fig. 3** *Il36rn−/−* mice are protected from diet-induced obesity and insulin resistance. **a–h** Male *Il36rn−/−* and wt mice of the same age were fed a HFD (60 kcal% fat) for 8–10 weeks starting from 8 weeks of age. Weights (**a**) and daily food intake per mouse (**b**) were monitored and data shown are pooled from three independent experiments (*wt n* = 15, *Il36rn−/− n* = 22). **c** EAT and SAT depots mass was measured after 10 weeks in HFD. **d, e** Intraperitoneal GTT on wt and *Il36rn−/−* mice after 8 weeks in HFD; glucose over time after i.p. glucose injection (**d**) and AUC (**e**) shown. **f, g** Intraperitoneal ITT on wt and *Il36rn−/−* mice after 9 weeks in HFD; glucose over time after insulin bolus-injected i.p. (**f**) and AUC (**g**) shown. **h** Fasting blood insulin levels for wt and *Il36rn−/−* mice after 10 weeks in HFD. **i** Relative protein expression levels of phospho-Akt (P-Akt)/total Akt (T-Akt) and phospho-insulin receptor β (P-IR)/total insulin receptor β (T-IR) in the liver, skeletal muscle and EAT of wt (*n* = 7) and *Il36rn−/−* (*n* = 4) mice after insulin injection, as indicated by densitometry analysis of western blot data (in Source Data file). **j–l** Protein expression of P-IR, T-IR, P-Akt and T-Akt in the liver (**j**), skeletal muscle (**k**) and EAT (**l**) of representative wt (*n* = 3) and *Il36rn−/−* (*n* = 3) mice after insulin injection. Data represent means ± SEM. Statistical analysis using two-tailed unpaired student's *t*-test for (**c**), (**g**) and (**i**), two-tailed Mann–Whitney test for (**b**), (**h**) and (**e**) and RM two-way ANOVA for (**a**), (**d**) and (**f**). ns $p > 0.05$, *$p < 0.05$, **$p < 0.01$, ***$p < 0.001$. Source data are provided as a Source Data file

*Il36rn* deficiency does not significantly alter the inflammatory profile of adipose tissue induced by exposure to HFD.

The IL-36 cytokine family has recently been shown to play a role in the homoeostasis of the intestinal mucosa where it can act opposingly to promote acute inflammation, as well as tissue homoeostasis and resolution[9,20,22,26]. Interestingly, we found that

gene expression levels of *Il36a* and *Il36b* were constitutively elevated in the colons of *Il36rn*−/− mice in the steady state. In addition, activation of the MAPK (p42/p44) signalling pathway was also found to be elevated in the colon, indicating enhanced IL-36 receptor signalling in the colonic tissue microenvironment in *Il36rn*−/− mice (Fig. 4a, b). In contrast, phosphorylation of

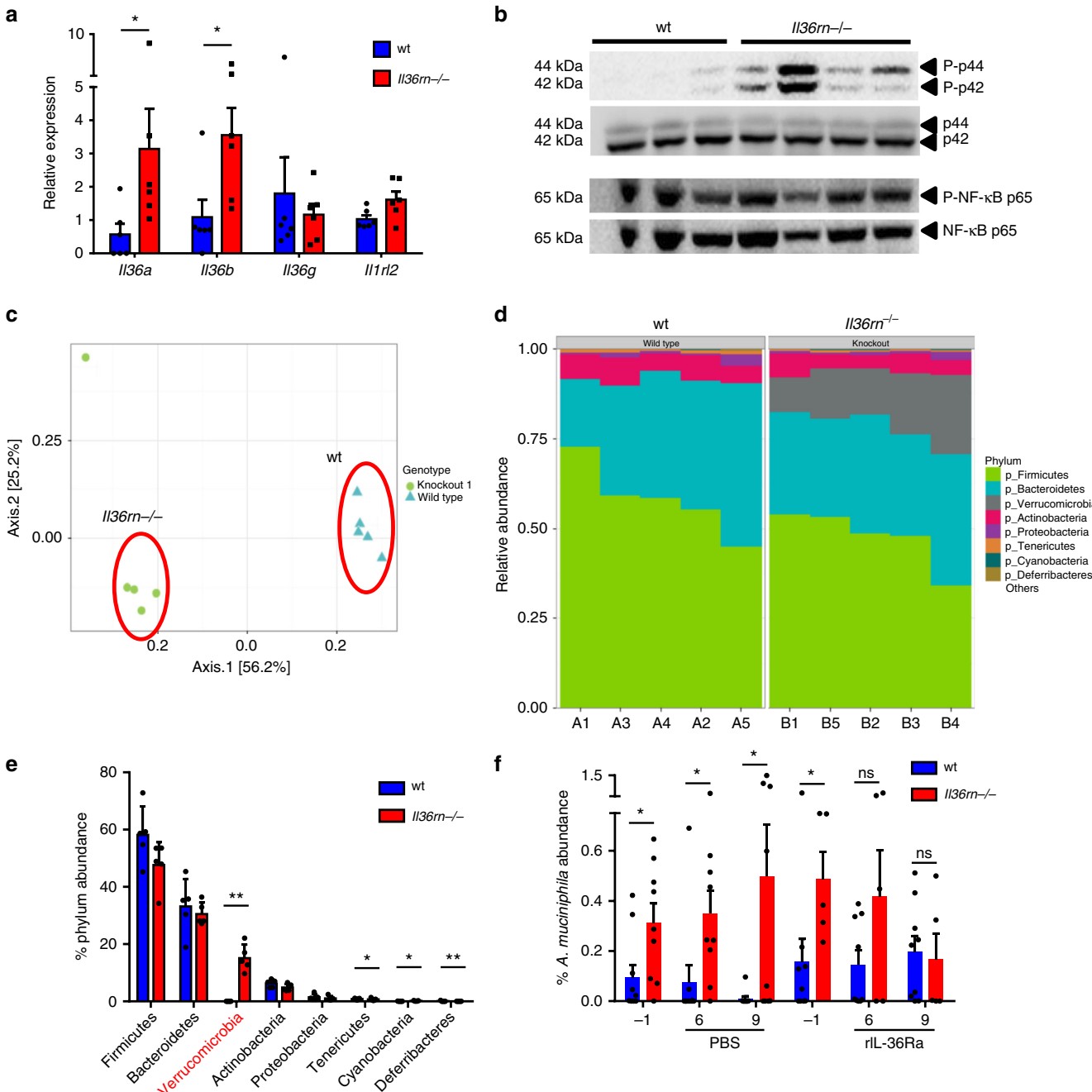

**Fig. 4** Altered composition of intestinal microbiome in *Il36rn*−/− mice dependent on IL-36 activity. **a** Relative gene expression of IL-36 cytokines and IL-36 receptor in the colon of wt and *Il36rn*−/− mice (*n* = 6 per group). **b** Protein expression of phospho-p44/p42 (Thr202/Tyr204, P-p44/P-p42) with total p44/p42, and phospho-NF-κB p65 (Ser536, P-NF-κB p65) with total NF-κB p65, in the colons of wt (*n* = 3) and *Il36rn*−/− (*n* = 4) mice. **c** Weighted ordination by principal coordinate analysis (PCoA) of beta-diversity of intestinal microbiome composition in wt and *Il36rn*−/− mice (*n* = 5 per group). **d** Proportional relative abundance of the most abundant phyla in the microbiome of wt and *Il36rn*−/− mice. **e** Percent relative abundance means ± SD of the eight most abundant phyla, Kruskal–Wallis rank sum test to determine significance. **f** Percent relative abundance of *Akkermansia muciniphila* in regard to total bacteria in the microbiome of 8–10-week-old wt and *Il36rn*−/− mice, as determined by qPCR using specific primers, before (day −1) and after (days 6, 9) three i.p. injections of rIL-36Ra (*n* = 9 and 5) or PBS control (*n* = 9) (days 1, 2, 3) were performed . Data show means ± SEM unless otherwise described. Statistical analysis using two-tailed Mann–Whitney test for (**a**) and (**f**), or as described. ns *p* > 0.05, *\**p* < 0.05, **\**p* < 0.01. Source data are provided as a Source Data file

**Table 1 OTUs identified at strain level in the microbiome of wt and *Il36rn−/−* mice**

| OTU, genus | Strain name | Change observed | Adj *P*-value |
|---|---|---|---|
| t__81483, g__94otu1084 | *Atopostipes suicloacalis* | Enriched in wild type | 0.0151 |
| t__284, g__*Corynebacterium* | *Corynebacterium ammoniagenes* | Enriched in wild type | 0.00354 |
| t__61727, g__*Brachybacterium* | *Brachybacterium sp.* Lact5.2 | Enriched in knockout | $6.59×10^{-08}$ |
| t__68330, g__*Bacteroides* | *Bacteroides acidifaciens* | Enriched in knockout | 0.00281 |
| t__43597, g__94otu20339 | *Alistipes shahii* WAL 8301 | Enriched in knockout | 0.0176 |
| t__40900, g__*Flexispira* | *Helicobacter mastomyrinus* | Enriched in wild type | $2.22×10^{-05}$ |
| t__78858, g__*Akkermansia* | *Akkermansia muciniphila* ATCC BAA-835 | Enriched in knockout | $1.08×10^{-58}$ |

the p65 subunit of NFκβ was found to be similarly high in both wt and *Il36rn−/−* colon tissues, suggesting constitutive activity of this pathway in the steady-state colon (Fig. 4b). The colonic gene expression levels of several other inflammatory mediators implicated in regulating intestinal inflammation and/or obesity and metabolic disease, including *Il1b*, *Tnfa*, *Il6*, *Il10* and *Ifng*, were similar between wt and *Il36rn−/−* mice (Supplementary Fig. 3a), indicating that loss of IL-36Ra expression did not lead to a broad non-specific state of inflammation in the gut. Furthermore, IL-6 and TNFα were also found to be expressed at similar levels in the serum of wt and *Il36rn−/−* mice after HFD, indicating that loss of IL-36Ra expression did not significantly alter systemic inflammation and contribute to the effects on weight gain and metabolism described (Supplementary Fig. 3b, c).

Prompted by these observations and the emerging important role for the intestinal microbiome in influencing the development of obesity[5,9], we investigated whether *Il36rn* deficiency resulted in any alterations in the constituents of the intestinal microbiome through 16S V4 rRNA gene sequencing. This analysis revealed significant beta-diversity differences in the overall composition of the intestinal microbiome between wt and *Il36rn−/−* mice (PERMANOVA, $p = 0.009$), with weighted ordination analysis based on Bray–Curtis dissimilarity values showing clear separation according to genotype (Fig. 4c). At the phylum level, the most striking difference observed was a significant increase in the abundance of *Verrucomicrobia* ($p = 0.01$, Kruskal–Wallis rank sum test), which was found to be >1600-fold more abundant in *Il36rn−/−* mice compared with wt (15.2 ± 4.65% vs. 0.0091 ± 0.0114% relative abundance, respectively) (Fig. 4d, e). Other phyla with significant, albeit minor, differences in abundance were *Cyanobacteria* (enriched in knockout), *Tenericutes* and *Deferribacteres* (enriched in wt) (Fig. 4d, e). A deeper analysis of differentially detected operational taxonomic units (OTU) at the genus/strain level, identified *Akkermansia muciniphila*, a member of the *Verrucomicrobia* phylum, as the most significantly enriched microbial strain among those identified in the intestinal microbiome of *Il36rn−/−* mice ($p = 1.08 × 10^{-58}$) (Table 1). Recent evidence indicates that *A. muciniphila* can play an important protective role against obesity and metabolic dysfunction in mice and may also play a similar role in humans[11,12,16]. The relative outgrowth of this strain was confirmed through qPCR of stool samples from an additional cohort of mice, using specific primers for *A. muciniphila*, and universal bacteria primers. Importantly, this relative outgrowth was found to be specifically dependent on *Il36rn* deficiency and was lost upon treatment of *Il36rn−/−* mice with recombinant IL-36Ra (Fig. 4f). Three intraperitoneal injections of IL-36Ra on days 1, 2 and 3, led to gradually decreasing abundance levels of *A. muciniphila*, similar to the levels observed in wt mice, as detected by qPCR on days 6 and 9 (Fig. 4f). These data demonstrate that loss of expression of the IL-36 receptor antagonist results in enhanced IL-36 cytokine gene expression in the colon and promotes the outgrowth of *A. muciniphila* in the intestinal microbiome.

**Increased mucus expression in the colons of *Il36rn−/−* mice.** We next sought to investigate what changes may be evident in the colons of *Il36rn−/−* mice, which might influence the composition of the intestinal microbiome, and in particular the relative outgrowth of *A. muciniphila*. *A. muciniphila* is a Gram-negative anaerobe which colonises the mucus layer of the gastrointestinal tract and degrades mucins as its predominant energy source[27]. Histological examination of the colon of *Il36rn−/−* mice through haematoxylin and eosin (H&E) and Alcian Blue/Periodic acid–Schiff (AB/PAS) staining, revealed a significant increase in the number of mucus-secreting goblet cells in the colon (Fig. 5a, c). This increase was also accompanied by a thickening of the outer mucus layer and significantly elevated expression of the *Muc2* gene (Fig. 5b, d, e), demonstrating that *Il36rn* deficiency leads to increased production of mucus in the colon providing an abundant source of nutrients to support the relative outgrowth of *A. muciniphila*. Consistent with these changes, we also observed improved intestinal barrier function in *Il36rn−/−* mice after HFD, as demonstrated by decreased gut permeability to FITC-dextran (Fig. 5f).

It has recently been reported that IL-36 can act to drive the expression of IL-9 by CD4[+] T cells in the colons of mice[21] and IL-9 has been previously reported to play a central role in promoting goblet cell hyperplasia in the gut[28]. Interestingly, levels of IL-9 secretion were found to be significantly increased in colonic tissue from *Il36rn−/−* mice compared with wild-type controls (Supplementary Fig. 4a). Secretion levels of IL-13 and IL-22, which have, also, been implicated in driving goblet cell differentiation[29,30], were not found to be significantly altered (Supplementary Fig. 4b, c).

To explore the mechanism through which enhanced IL-36 activity leads to increased mucus secretion in the colon, we investigated whether administration of rIL-36Ra, which inhibited the relative outgrowth of *A. muciniphila* (Fig. 4f), could also reverse these changes. Interestingly, *Il36rn−/−* mice treated with rIL-36Ra had decreased goblet cell numbers, that were comparable with those of the wild-type mice (Fig. 6a, b). In contrast, treatment with an anti-IL-9-neutralizing antibody did not alter goblet cell numbers in *Il36rn−/−* mice, suggesting that elevated IL-9 expression does not play a significant role (Supplementary Fig. 4d, e). Furthermore, ex vivo stimulation of wild-type colon explants with IL-36α for 4 h led to a significant increase in *Muc2* gene expression, suggesting that IL-36 cytokines can act directly to stimulate increased mucus production (Fig. 6c). Together, these data confirm that IL-36 cytokines can alter the colonic tissue microenvironment to promote the relative outgrowth of *A. muciniphila*.

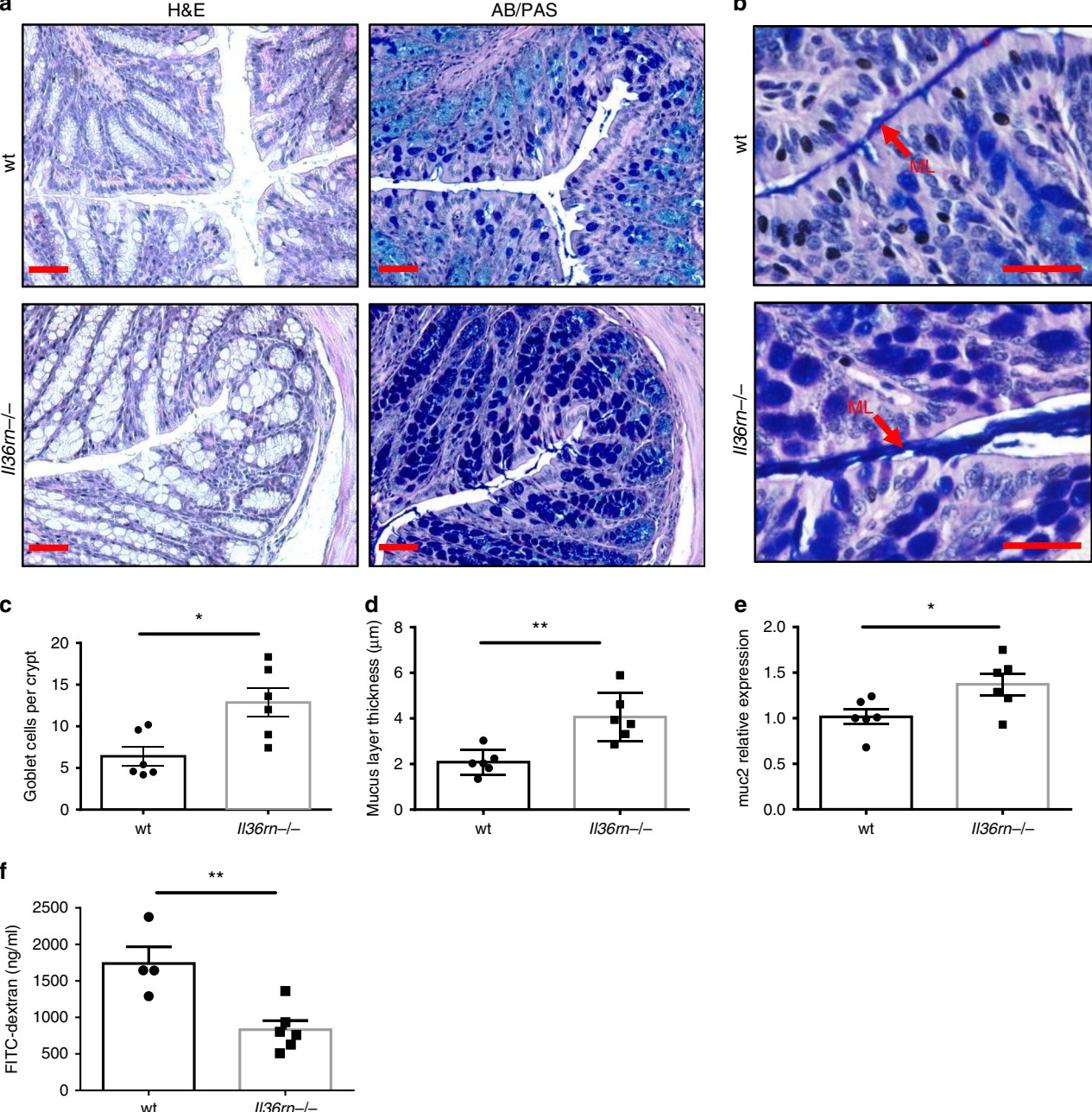

**Fig. 5** Increased colon mucus levels and reduced intestinal permeability in *Il36rn−/−* mice. **a** Representative micrographs of distal colon sections from wt and *Il36rn−/−* mice stained with H&E and AB/PAS. Scale bar = 1 μm. **b** Representative micrographs of colon sections stained with AB/PAS with mucus layer (ML) indicated by red arrow. Scale bar = 20 μm. **c, d** Quantitation of average goblet cells per crypt (**c**) and mucus layer thickness (**d**) in distal colon sections (n = 6 per group, 10 crypts measured per sample). Data shown representative of three independent experiments. **e** Relative gene expression of *Muc2* gene in the distal colon of wt and *Il36rn−/−* mice (n = 6 per group). **f** Fluorescein isothiocyanate (FITC)-dextran concentration in the serum of wt and *Il36rn−/−* mice on HFD for 12 weeks, 4 h after oral gavage (n = 4 and 6 per group). Data show means ± SEM. Statistical analysis using two-tailed unpaired student's *t*-test. *$p < 0.05$, **$p < 0.01$. Source data are provided as a Source Data file

**The microbiome influences reduced obesity in *Il36rn−/−* mice**. In an effort to examine the influence of the altered intestinal microbiome described above on the reduced weight gain and metabolic dysfunction observed in the setting of *Il36rn* deficiency, we sought to equilibrate the gut microbiome through prolonged (8 weeks) cohousing of *Il36rn−/−* and wt mice in the same cage, to investigate whether this altered the response to HFD exposure. As previously reported in similar studies, cohousing of mice resulted in a relative equalization of the constituents of the intestinal microbiome[31,32], as determined by analysis of beta-diversity when compared with separately housed mice (PER-MANOVA, $p = 0.001$). Weighted ordination analysis demonstrated similar positional clustering for all cohoused mice in contrast to separately housed mice which clustered according to genotype (Fig. 7a). Critically, this equalisation coincided with a complete loss of the relative outgrowth of the phylum *Verrucomicrobia* ($0.149 \pm 0.0367\%$ relative abundance for *Il36rn−/−*, $0.183 \pm 0.133\%$ for wt), and consequently *A. muciniphila*, as

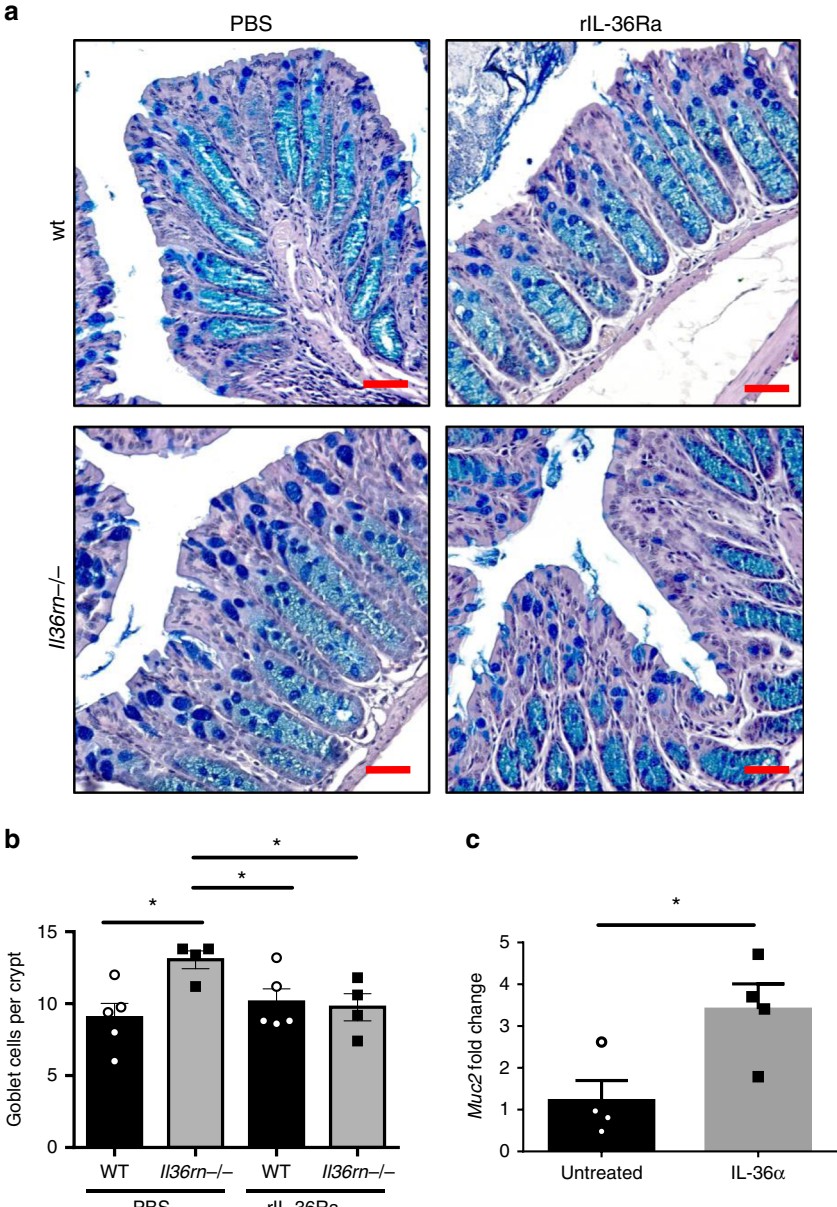

**Fig. 6** IL-36 signalling increases goblet cell numbers and *Muc2* gene expression. **a, b** Representative micrographs of AB/PAS-stained sections (**a**) and average goblet cells per crypt quantitation (**b**) in the colon of wt and *Il36rn−/−* mice after (day 6) three consecutive injections (days 1, 2, 3) of rIL-36Ra or PBS control ($n = 4$ and 5 per group). Scale bar = 1 μm. **c** Relative *Muc2* gene expression in wt colon explants either untreated or stimulated for 4 h with recombinant IL-36α (200 ng/ml). Data show means ± SEM. Statistical analysis using two-tailed unpaired student's *t*-test. *$p < 0.05$. Source data are provided as a Source Data file

demonstrated through relative OTU analysis and confirmed by qPCR for *A. muciniphila* detection (Fig. 7b–d). Cohoused *Il36rn−/−* mice exhibited a significant enrichment of *Bacteroidetes*, and decreased abundance of *Actinobacteria* when compared with their cohoused wt counterparts, but none of the previously differentially abundant phyla were found to be altered in cohoused mice (Fig. 7b, c).

Strikingly, cohoused *Il36rn−/−* mice now exhibited similar levels of weight gain, adipose tissue mass and glucose and insulin intolerance when compared with their wild-type counterparts upon exposure to HFD for 8–10 weeks (Fig. 7e–j). These data demonstrate a clear association between the altered intestinal microbiome observed in *Il36rn−/−* mice and their protection from obesity and metabolic disease.

## Discussion

Cytokines of the interleukin-1 family are emerging as one of the most important mediators of obesity and metabolic health, with diverse and often opposing roles in either driving inflammatory mechanisms associated with disease pathogenesis, e.g. IL-1β, or inhibiting inflammation and/or caloric intake to promote the maintenance of metabolic health, e.g. IL-33, IL-18 and IL-37[17,24]. Although the precise mechanisms through which members of the IL-1 family exert these dichotomous effects are incompletely understood, to date all members have been investigated for their roles in disease with the exception of the IL-36 subfamily.

Similar to related IL-1 family members, expression of IL-36 cytokines, specifically IL-36γ, is elevated in the serum of obese patients. Importantly, these elevated levels are negatively

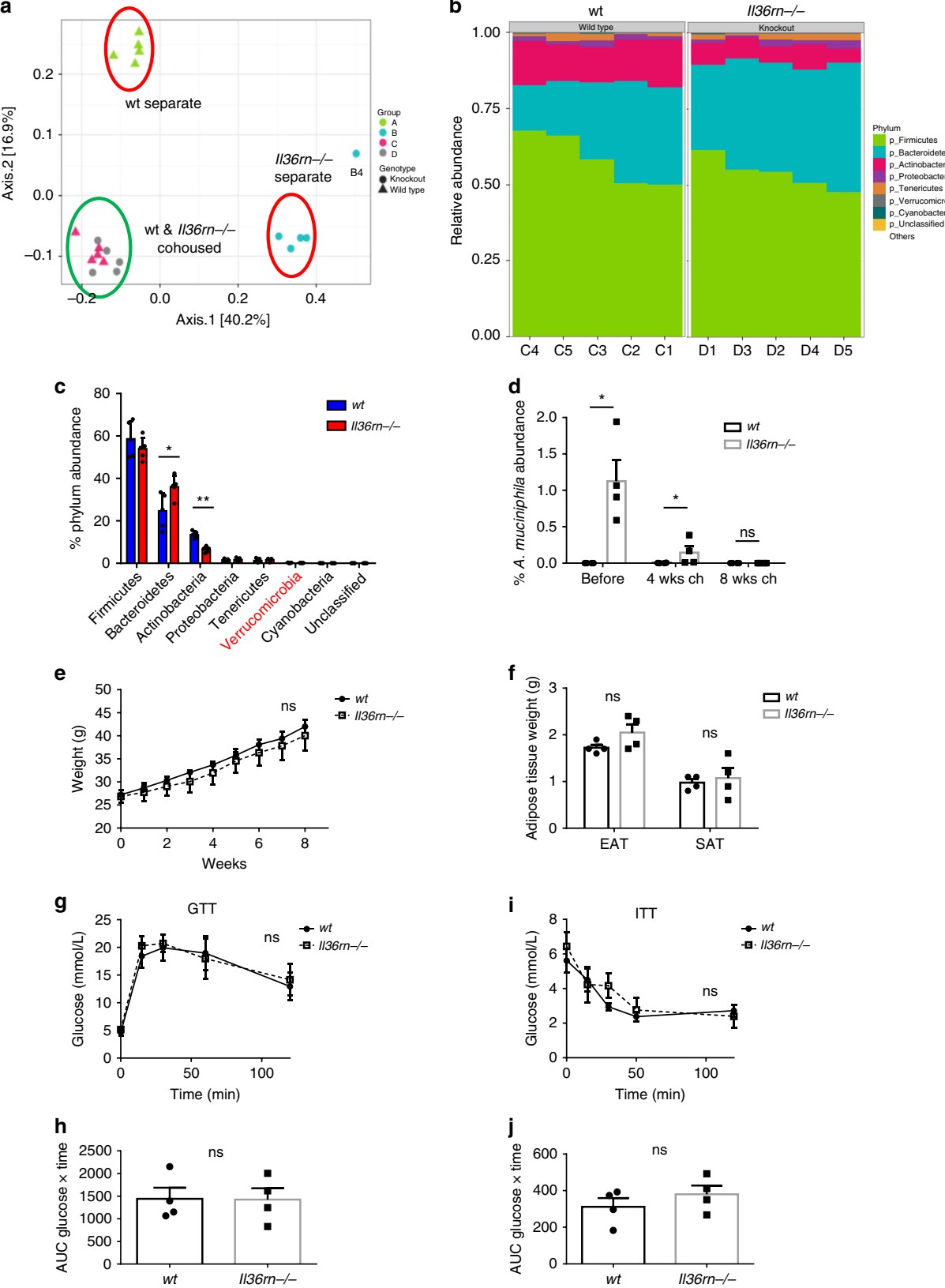

correlated with primary indicators of metabolic health, i.e. HbA1c and FBG levels, among obese patients presenting with diabetes, indicating that elevated IL-36 cytokines may play a protective role in reducing blood sugar levels among obese patients who have developed consequent metabolic disease.

In an effort to explore mechanistically how the IL-36 family might regulate obesity and metabolic disease, we investigated what role, if any, these cytokines might play in disease pathogenesis in mice with a deficiency in the IL-36 receptor antagonist (*Il36rn*−/−) and the IL-36 receptor (*Il1rl2*−/−). This analysis

**Fig. 7** Microbiome analysis and obesity phenotype of cohoused wt and *Il36rn*−/− mice. **a** Weighted ordination analysis of beta-diversity of intestinal microbiome composition in cohoused for 8 weeks and separately housed wt and *Il36rn*−/− mice (*n* = 5 per group). **b** Proportional relative abundance of the most abundant phyla in the microbiome of cohoused wt and *Il36rn*−/− mice. **c** Percent relative abundance means ± SD of the eight most abundant phyla (including unclassified phylum), Kruskal–Wallis rank sum test to determine significance. **d** Percent relative abundance of *A. muciniphila* in regard to total bacteria before and after cohousing wt and *Il36rn*−/− mice for 4 and 8 weeks, as determined by qPCR (*n* = 4 per group). **e–h** Aged matched male wt and *Il36rn*−/− mice were cohoused since weaning in the same cage for 8 weeks (*n* = 4 per group). They were then given HFD for 8–10 weeks, and weights were monitored weekly (**e**). **f** EAT and SAT depots mass after 10 weeks in HFD. **g**, **h** GTT on cohoused mice after 8 weeks in HFD, glucose over time (**g**) and AUC (**h**) shown. **i–j** ITT on cohoused mice after 9 weeks in HFD, glucose over time (**i**) and inverted AUC (**j**) shown. Data show means ± SEM unless otherwise described. Statistical analysis using two-tailed Mann–Whitney test for **d**, two-tailed unpaired student's *t*-test for (**f**), (**h**) and (**j**), RM two-way ANOVA for (**e**), (**g**) and (**i**) or as described. ns *p* > 0.05, *\*p* < 0.05, *\*\*p* < 0.01. Source data are provided as a Source Data file

revealed that constitutively hyperactive IL-36 cytokine activity (*Il36rn*−/−) led to a significant reduction in weight gain and adiposity, with improved glucose and insulin tolerance, without altering food intake. These data demonstrate that elevated IL-36 family signalling plays a broadly similar protective role to related IL-1 family members IL-18, IL-33 and IL-37[33–36]. In contrast to what has been reported under settings of IL-18R or IL-33R deficiency[33,37], we observed that mice with a deficiency of the IL-36 receptor (*Il1rl2*−/−) did not display any changes in adiposity development after HFD, demonstrating that constitutive IL-36R expression does not influence disease progression in this model and suggesting that protection from disease is only evident under conditions of elevated IL-36 family activity. In this regard, it is interesting to note that elevated levels of IL-36α, β and γ are found in the serum of a number of lean as well as obese individuals (Fig. 1), and it would be of interest to investigate whether these patients are less susceptible to the development of obesity and metabolic dysfunction.

Related IL-1 family members have previously been shown to influence the development of insulin resistance, at least in part, through altering the nature of adipose tissue inflammation[17]. However, we did not observe any significant alterations in immune cell infiltration in the adipose tissues of *Il36rn*−/− mice, suggesting that this cytokine family may regulate obesity development at a different tissue site. IL-36 cytokines are well established as mediators of inflammation in psoriatic skin, and more recently, as important regulators of homoeostasis at the gut mucosa[18]. Prompted by these discoveries, and the emerging role of the intestinal microbiome in regulating metabolic health, we investigated whether IL-36 cytokines can alter the composition of the intestinal microbiome, and identified a significant outgrowth of the mucin-degrading bacterial strain *Akkermansia muciniphila* in *Il36rn*−/− mice. *A. muciniphila* has recently emerged as an important commensal which can protect against obesity and metabolic disease in mice and has been associated with improved health among obese patients[11,12,16]. Although the precise host mechanisms through which the bacterium exerts these effects are yet to be fully identified, our mouse data demonstrate that its abundance is regulated by IL-36 family cytokines and indicate that it plays a central role in mediating protection from disease in *Il36rn*−/− mice. While alterations in the intestinal microbiome have previously been described in studies using mice deficient in related IL-1 family members, the significance of these findings in relation to obesity and metabolic health has not been reported. Indeed, two recent studies have found that IL-33 and IL-18, which suppress obesity and metabolic dysfunction, can in fact act to limit *A. muciniphila* abundance in mice under certain settings[32,38]. This suggests that IL-36 cytokines act in a mechanistically distinct fashion to protect from disease. In line with this observation, while IL-18 has been reported to act to suppress food intake in mice[33], no such changes were observed in *Il36rn*−/− mice in this study. It is also intriguing to note that IL-18 has been reported to suppress the differentiation of mucus-

secreting goblet cells[31], while *Il36rn* deficiency leads to increased goblet cell numbers as well as increased mucus expression and likely creates an environmental niche to support *A. muciniphila* outgrowth[17,27,39]. Similarly, numerous studies have uncovered prominent roles for both IL-18 and IL-33 in regulating the homoeostasis of adipose tissue- resident immune cells, while no such changes were found in *Il36rn*−/− mice in this study[17].

While demonstrating novel protective effects against disease, this study does not rule out a significant role for IL-36 cytokines in having further, unidentified important mechanistic roles in mediating these effects. In this regard, our observation that higher IL-36γ levels are inversely associated with blood glucose levels among diabetic patients is of particular significance and it would be of interest to determine whether IL-36 cytokines can act specifically to regulate glucose secretion/uptake among obese patients. Similarly, it has previously been reported that IL-36 cytokines can act to suppress peroxisome proliferator-activated receptor γ and perhaps restrict adipocyte differentiation in vitro[40]. A further important consideration is the fact that several of the obese diabetic patients included in our study were under treatment with metformin which has previously also been found to alter the composition of the intestinal microbiome in diabetic patients and specifically increase the abundance of *A. muciniphila* in the intestines of mice[14,15]. Therefore, a deeper investigation of the effects of IL-36 activity in obese diabetic patients is warranted.

IL-36 cytokines have been described as playing dichotomous pro-inflammatory and pro-resolving roles across several tissue sites, including the gut[18]. In this regard, it is possible that a constitutive basal level of inflammation in *Il36rn*−/− mice may play a confounding role in the phenotype described. However, it is noteworthy that we were unable to detect any significant changes in expression of inflammatory mediators implicated in obesity-dependent metabolic disease, either in colonic tissue in the steady state, or systemically in the serum after HFD exposure (Supplementary Fig. 3).

In summary, our data provide novel insight into how IL-36 cytokines can act to regulate the pathogenesis of obesity and metabolic disease and demonstrate for the first time their ability to direct the composition of the intestinal microbiome towards a more metabolically healthy state. Furthermore, these data provide important new advances in understanding the dichotomous nature of the broader IL-1 family as critical regulators of inflammatory disease and identify the role of the intestinal microbiome as an important regulator of these effects.

## Methods

**Human subjects.** Serum samples were obtained from 104 adult donors, of whom 67 were classified as obese having a body mass index of ≥30 kg/m². A clinical assessment of all subjects was undertaken upon enrolment, and parameters including sex, age, BMI, weight in kg, HbA1c, FBG, cholesterol, triglycerides, LDL and HDL, as well as current medication use was recorded (Supplementary Table 1). Levels of human IL-36 alpha, IL-36 beta, IL-36 gamma and IL-36Ra were measured in the sera of patients using DuoSet ELISA kits (R&D Systems, Minneapolis, MN) according to the

manufacturer's instructions, in a blinded fashion. For further analysis, obese subjects were classified as type 2 diabetic using HbA1c ≥48 mmol/mol as per WHO and ADA guidelines[25] or non-diabetic with HbA1c <48 mmol/mol. Patients with controlled diabetes, i.e. HbA1c <48 mmol/mol, undergoing current antidiabetic therapy were excluded from the study. Ethical approval for the human study was obtained from the Ethics Committees at St. Vincent's University Hospital Dublin, and all patients and control subjects gave written, informed consent.

**Mice.** Wild-type (wt) and *Il36rn−/−* mice on C57BL/6 background were previously described[19] and bred in-house. *Il36rn−/−* mice were rederived from heterozygotes in our facility. *Il1rl2−/−* mice on C57BL/6 background obtained from Amgen under Material Transfer Agreement were described previously[22]. Male mice were fed irradiated normal chow diet and weights were measured monthly from 8 weeks of age; food and water intake was measured weekly. For diet-induced obesity experiments, three independent cohorts of male mice were fed with a HFD (60 kcal% fat, D12492i, Research Diets Inc.) for 8–12 weeks starting from 8 to 10 weeks of age, and weights and food intake were measured weekly. For cohousing experiments, wt and *Il36rn−/−* mice were housed in the same cage at a 1:1 ratio since weaning, for 8 weeks, before being given HFD for 8–10 weeks, their weights measured weekly. To determine specificity of IL-36Ra effects on microbiome and goblet cells, wt and *Il36rn−/−* mice were injected i.p. with recombinant IL-36Ra (R&D Systems) (1.5 µg/mouse) on 3 consecutive days (days 1, 2, 3) and colon tissue collected in formalin on day 6 for histology, or faecal pellets collected for DNA extraction and bacteria detection using qPCR on days −1, 6 and 9. For IL-9 neutralisation experiments, wt and *Il36rn−/−* mice (n = 6–8 per group) were administered by i.p. injection either with 100 µg/mouse anti-mouse IL-9 monoclonal antibody (InVivoMAb, clone 9C1, BioXCell) or isotype control (InVivoMab mouse IgG2a isotype control, clone C1.18.4, BioXCell) every other day for 5 days and colons were subsequently harvested for goblet cell enumeration. All animal experiments were performed with ethical approval by TCD Animal Research Ethics Committee and under license by the Irish Health Products Regulatory Authority (project authorization no: AE19136/PO77).

**Glucose and insulin tolerance tests.** For glucose tolerance test, mice were injected i.p. with 2 g/kg glucose (Sigma) after overnight fasting. For insulin tolerance test, they were injected i.p. with 1 U/kg insulin (recombinant human, Sigma) after overnight fasting. Blood glucose was measured by submandibular bleeding at defined time intervals, using handheld blood glucose monitor (On Call Vivid, ACON Laboratories or FreeStyle Lite, Abbot Laboratories).

**RNA extraction and real-time quantitative RT-PCR.** Colon samples from wt and *Il36rn−/−* mice were harvested and stored in RNA*later* (Sigma) at −20 °C. Total RNA was extracted using Isolate II RNA Mini Kit (Bioline, London, UK) after tissue disruption using BeadBug homogenizer and 1.5 mm Zirconium beads (Benchmark Scientific, Inc.). Reverse transcription was performed using High-Capacity cDNA Kit (Applied Biosystems, Foster City, CA) following the manufacturer's instructions, with 100 ng of total RNA per sample. Real-time PCR for the indicated transcripts was performed in triplicate using specific TaqMan Gene Expression Assays (Supplementary Table 2) and TaqMan Fast Universal PCR Master Mix in a 7900HT Fast Real-Time PCR System (Applied Biosystems, Foster City, CA). 18S ribosomal RNA was used for normalisation and relative expression levels were calculated using the ΔΔCt method.

**16S V4 rRNA gene sequencing for microbiome profiling.** Intestinal microbiome profiling analysis was performed by Second Genome Inc. (CA, USA). DNA isolation from 10-week-old mouse fecal material was performed with the MoBio PowerMag® Microbiome kit (Carlsbad, CA) according to the manufacturer's guidelines and optimised for high-throughput processing. All samples were quantified via the Qubit® Quant-iT dsDNA High Sensitivity Kit (Invitrogen, Life Technologies, Grand Island, NY) to ensure that they met minimum concentration and mass of DNA. To enrich the sample for bacterial 16S V4 rDNA region, DNA was amplified utilising fusion primers designed against the surrounding conserved regions which are tailed with sequences to incorporate Illumina (San Diego, CA) adapters and indexing barcodes. Each sample was PCR amplified with two differently barcoded V4 fusion primers. Samples that met the post-PCR quantification minimum were advanced for pooling and sequencing. For each sample, amplified products were concentrated using a solid-phase reversible immobilisation method for the purification of PCR products and quantified by qPCR. A pool containing 16S V4 enriched, amplified, barcoded samples were loaded into a MiSeq® reagent cartridge, and then onto the instrument along with the flow cell. After cluster formation on the MiSeq instrument, the amplicons were sequenced for 250 cycles with custom primers designed for paired-end sequencing.

Second Genome's analysis software package was used for statistical analysis. Sequenced paired-end reads were merged using USEARCH and the resulting sequences were compared with an in-house strain database using USEARCH (usearch_global). All sequences hitting a unique strain with an identity ≥99% were assigned a strain Operation Taxonomic Unit (OTU). To ensure specificity of the strain hits, a difference of ≥0.25% between the identity of the best hit and the second-best hit was required (e.g. 99.75 vs. 99.5). For each strain OTU, one of the matching reads was selected as representative and all sequences were mapped by USEARCH (usearch_global) against the strain OTU representatives to calculate strain abundances. The remaining non-strain sequences were quality filtered and dereplicated with USEARCH. The resulting unique sequences were then clustered at 97% by UPARSE (de novo OTU clustering) and a representative consensus sequence per de novo OTU was determined. The UPARSE clustering algorithm comprises a chimera filtering and discards likely chimeric OTUs. All non-strain sequences that passed the quality filtering were mapped to the representative consensus sequences to generate an abundance table for de novo OTUs. Representative OTU sequences were assigned taxonomic classification via mothur's bayesian classifier, trained against the Greengenes reference database of 16S rRNA gene sequences clustered at 99%.

For beta-diversity analysis, abundance-weighted sample pairwise differences were calculated using the Bray–Curtis dissimilarity. Principal coordinate analysis (PCoA) was used to plot two-dimensional ordination. PCoA uses the sample-to-sample dissimilarity values (beta-diversity) to position the points relative to each other by maximising the linear correlation between the dissimilarity values and the plot distances. Permutational analysis of variance (PERMANOVA) was utilised for whole-microbiome significance testing of beta-diversity differences. For taxon significance testing, univariate differential abundance of OTUs was tested using a negative binomial noise model for the overdispersion and Poisson process intrinsic to these data, as implemented in the DESeq2 package[41], and described for microbiome applications[42]. It takes into account both technical and biological variability between experimental conditions. DESeq was run under default settings and q-values were calculated with the Benjamini–Hochberg procedure to correct p-values, controlling for false discovery rates.

**Bacteria DNA extraction and quantitative PCR.** DNA was extracted from mouse faecal material after homogenisation with 1.5 mm zirconium beads using BeadBug homogenizer (Benchmark Scientific, Inc.). DNA isolation was performed using QiaAmp Fast DNA Stool Mini Kit (QIAGEN) following the manufacturer's protocol. An amount of 10–20 ng of DNA was then used in qPCR reactions for *A. muciniphila* with specific primers (AM1: CAGCACGTGAAGGTGGGGAC, AM2: CCTTGCGGTTGGCTTCAGAT), and for total bacteria (UniF340: ACTCCTAC GGGAGGCAGCAGT, UniR514: ATTACCGCGGCTGCTGGC) for normalisation. qPCR was performed in 7900HT Fast Real-Time PCR system using Fast SYBR Green Master Mix (Applied Biosystems, Foster City, CA).

**Histology and staining.** Distal colon samples were fixed in 10% neutral buffered formalin (Medical Supply, Mulhuddart, Dublin), or Carnoy's solution for mucus layer analysis, and embedded in paraffin. Blocks were sectioned in a microtome and 5-µm thickness sections were mounted in Superfrost Plus adhesion slides (Thermo Scientific, Braunschweig, Germany). Slides were stained with H&E and AB/PAS for histological assessment and to enumerate goblet cells per crypt and mucus layer thickness in the tissue.

**Cytokines and insulin ELISA assays.** Colon explants were cultured for 24 h (37 °C, 5% CO₂) to measure secretion of cytokines IL-9, IL-13 and IL-22 in the supernatants using mouse uncoated ELISA kits (Invitrogen, Ready-SET-Go! ELISA kits, eBioscience), or for 4 h in the presence of 200 ng/ml IL-36α to measure *Muc2* mRNA expression. IL-6 and TNF-α were measured in the serum of mice after HFD using mouse uncoated ELISA kits (Invitrogen, Ready-SET-Go! ELISA kits, eBioscience). Levels of fasting plasma insulin were measured using Ultra Sensitive Mouse Insulin ELISA Kit (Crystal Chem) according to the manufacturer's instructions.

**Flow cytometry.** Adipose tissue was digested with 1 mg/ml collagenase D, passed through a cell strainer and red blood cells were lysed. Stromal vascular cells were then stained for flow cytometric analysis using fluorophore-conjugated antibodies specific for mouse CD45 (30-F11), F4/80 (BM8), CD11b (M1/70), CD11c (N418) (eBioscience, Invitrogen) and CD301 (LOM-14) (BioLegend), and Aqua Live/Dead stain (Invitrogen) to identify live-cell population. LSR Fortessa instrument (BD Biosciences) was used and data were analysed using FlowJo software (TreeStar). Cell enumeration was performed using CountBright Absolute Counting Beads for flow cytometry (Invitrogen).

**Western blotting.** For insulin sensitivity assays, protein lysates were prepared from liver, leg skeletal muscle and epididymal white adipose tissue harvested from mice fed with a HFD for 8 weeks, 10 min after intraperitoneal challenge with 1.5 U/kg insulin (recombinant human, Sigma), using RIPA Lysis Buffer System (Santa Cruz). For analysis of signalling pathways downstream of IL-36 in colon tissues, tissues were harvested from wt and *Il36rn−/−* mice and lysed in RIPA buffer. Protein concentration was determined with Bicinchoninic Acid Kit for Protein Determination (Sigma) and lysates were diluted with NuPAGE LDS Sample Buffer and Reducing Agent (ThermoFisher Scientific) and heated for 10 min at 70 °C. Samples were loaded on precast NuPAGE Novex 4–12% Bis-Tris gradient gels (Invitrogen) or RunBlue Bis-Tris 4–12% gels (Expedeon) and SDS-PAGE was performed in XCell *SureLock* Mini-Cell Electrophoresis system (ThermoFisher Scientific) for 120 min at 100 V. Separated proteins were transferred to Invitrolon

polyvinylidene fluoride (PVDF) membranes (ThermoFisher Scientific) for 1 h at 30 V, followed by blocking with 5% BSA or skimmed milk and incubation with appropriate antibodies. Primary antibodies used were Phospho-NF-κB p65 (Ser536) (93H1) Rabbit mAb (1:1000, Cell Signaling), NF-κB p65 (C-20) Rabbit polyclonal (1:1000, Santa Cruz), Phospho-AKT (S473) (D9E) XP Rabbit mAb (1:1000, Cell Signaling), Akt (pan) (C67E7) Rabbit mAb (1:1000, Cell Signaling), Phospho-p44/42 MAPK (Thr202/Tyr204) (D13.14.4E) XP Rabbit mAb (1:1000, Cell Signaling), p44/42 MAPK (Erk1/2) (137F5) Rabbit mAb (1:1000, Cell Signaling), Phospho-IGF-I Receptor β (Tyr1135/1136)/Insulin Receptor β (Tyr1150/1151) (19H7) Rabbit mAb (1:1000, Cell Signaling) and Insulin Receptor β (4B8) Rabbit mAb (1:1000, Cell Signaling). Secondary antibody used was Goat anti-Rabbit IgG (1:5000, Sigma) horseradish peroxidase conjugated. Blots were developed in FUSION-FX chemiluminescence system (Vilber) using SuperSignal™ West Pico PLUS Chemiluminescent Substrate (ThermoFisher Scientific). Densitometry analysis of immunoblots was performed using ImageJ software. Uncropped versions of blots can be found in the Source Data file.

**FITC-dextran for intestinal permeability**. Mice on a HFD for 12 weeks were administered 20 mg/mouse FITC-dextran (Sigma) by oral gavage after overnight fasting. After 4 h, blood was collected and serum was prepared. Serum was diluted 1/2 in PBS and fluorescence at 490/525 nm was measured, using the standard curve of FITC-dextran and serum from control (non-gavaged) animal as blank to determine FITC-dextran concentrations in serum.

**Statistical analysis**. Sample size for patient's serum analysis and mouse studies were chosen based on our published studies[35,43]. Analysis of cytokines in patients serum was carried out in a blinded fashion and subsequently segregated based upon clinical parameters. Analysis of data from mouse studies, i.e. goblet cell enumeration was carried out in a blinded fashion. Statistical analysis of individual data sets was performed by first analysing data sets for normal distribution (Shapiro–Wilk and Kolmogorov–Smirnov test) and equality of variance (F test), followed by unpaired two-tailed student's $t$-test or Mann–Whitney test for non-parametric data, as appropriate. For repeated measures analysis, RM two-way ANOVA with Bonferroni post hoc test was used instead. Correlation analysis were performed using Spearman's correlation test. All analysis and graph representation were performed using GraphPad Prism 6 software. Data are shown as means ± SEM unless specified otherwise. All qPCR data were analysed using the ΔΔCt method. Statistical details for each figure can be found in the figure legends.

## Data availability

All relevant data are available from the authors. Source data for all relevant figures provided as a Source Data file. 16S sequence data are available through GenBank, Accession Numbers MN211558–MN212548.

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

## Acknowledgements

This work was supported by grant funding from the National Children's Research Centre and the Health Research Board, Ireland (HRA-POR-2015-1066) to P.T.W. and a Swiss National Science Foundation grant (310030_163443/1) to M.K.

## Author contributions

E.G. and Y.E.H.S. contributed equally to this work. E.G., Y.E.H.S., K.M. and E.H. performed experiments and analysed the data. S.L.D. and P.F. provided reagents, designed experiments and analysed the data. M.K. provided reagents. D.O.S., A.M. and A.E.H. performed clinical study and analysed clinical data. E.G., Y.E.H.S and P.T.W. wrote the paper. P.T.W. conceived and directed the study.

## Additional information

**Competing interests:** The authors declare no competing interests.

**Peer review information:** *Nature Communications* would like to thank Hsin-Jung Joyce Wu and other, anonymous, reviewers for their contirbutions to the peer review of the is work. Peer review reports are available.

