## [Peer Review File · Nature Communications]

Reviewers' comments:

Reviewer #1 (Remarks to the Author):

The authors set to elucidate the role in obesity and metabolic disorder of the IL-36 family cytokines, a subgroup of the broader IL-1 family, which has been widely implicated in metabolic disease. The authors first observe that this class of molecules is increased in the sera of obese patients, and that its levels are negatively correlated with blood glucose levels in diabetic patients. They next observe that mice with a genetic suppression of the IL-36 signalling inhibitor (Il36rn KO) are protected from the age- and high-fat diet-induced metabolic disruptions. These effects appear to be mediated by an increased abundance of *A. muciniphila* in the Il36rn KO mice, which could be due to functional changes in the colon that promote the growth of the mucin-dependent bacterial species. Cohousing of WT and KO mice resulted in a normalization of the KO mice microbiota and a loss of metabolic improvements.

The paper is interesting and sheds light on a class of molecules underexplored in context of metabolic diseases and microbiota. However, in my view there are several controls that are missing, a better description of the glucose phenotype is needed, and the evidence concerning the mechanism is too descriptive.

Main points:

- What are the organs responsible for the improved glucose phenotype in the Il36rn KO mice? Hyperinsulinemic-euglycemic clamp, or at least a 2-deoxyglucose tracer studies coupled with pAKT measurements could help elucidating this. The insulin sensitivity should be also investigated in the mice on chow diet (shown in Figure 2).
- To support the claims that the elevated IL36 levels and signalling are important in regulating the metabolic homeostasis, the IL36 alpha, beta and gamma need to be supplemented alone, or in combination in the WT and in the KO mice. This is a critical experiment that could also suggest therapeutic relevance of these findings.
- Authors dedicate substantial effort in speculating the mechanisms related to the IL36, but this is not sufficiently addressed experimentally. Does supplementing WT mice with IL-9 promote a KO-like phenotype on the abundance of *Akkermansia*? Or conversely, does CD4+ depletion in the KO mice reduce mucin (and therefore *A. muciniphila*'s) levels?
- The Verucomicrobia in the WT mice shown in Fig. 4c is almost not present. This is surprising, and is not consistent with the literature. These inconsistencies might explain the remarkable increase of the Verucomicrobia abundance (> 1600 fold) in the KO mice. These results need to be confirmed in an independent groups of mice (and preferably also at different ages). The interpretation of these results is difficult without specifying the exact age of the mice – and this should be done in all the figures throughout the study.
- The colon mucin layer should be quantified in the WT vs. the KO mice.

Additional points:

- Authors need to confirm that the Il36rn KO mice indeed show increased IL36 downstream signalling.
- The study suggests that there is no difference of the macrophage subsets per gram tissue. Notably, despite the lack of calculated significance, in some cases there more than a double fold difference. Based on what power calculation were these animal numbers chosen? Each value should be presented as a data point, and I suggest increasing the animal number (coupled to power calculations) to allow proper interpretation of these results.
- In relation to the above comments - there is a large discrepancy between the levels of *A. muciniphila* detected in Fig 4D,E with the ones in figure 6D. Is there an error in this figure?

Reviewer #2 (Remarks to the Author):

This study identified that IL-36 may protect against metabolic dysfunction in both human and mice. Mechanistically, the authors suggested that the protective effects are dependent on the metabolically protective bacteria *A. muciniphila*. Overall, this manuscript reported an interesting and novel host-microbiota interaction that promotes metabolic health. However, there are some moderate concerns that the authors could address.

1. In Fig. 4E, the *A. muciniphila* level at the various time points in WT and IL-36^{rn-/-} mice without IL-36Ra treatment should be provided as controls. This is particularly important as rIL-36Ra injections seemed to bring up the *A. muciniphila* level in WT mice (the opposite of what you would expect from their results).
2. Fig. 6 co-housing result provided a strong positive association of *A. muciniphila* level and the metabolically protective phenotype. To prove a causal effect of *A. muciniphila*, the authors could gavage mice with *A. muciniphila* and examine whether this approach can produce protective phenotype. This would be the most convincing data for their claim.
3. IL-36 is classically associated with inflammatory responses. The authors should examine the potential tissue inflammation especially in HFD group to make sure this does not give confounding effect to their findings.

Response to Reviewers' comments:

We thank the reviewers for their insightful and helpful comments and suggestions. Based upon their recommendations, we have undertaken considerable additional experimentation which we believe significantly improves our study through a deeper mechanistic understanding of how IL-36 alters the colonic microenvironment to facilitate protection from disease, inclusion of improved control experiments, and an analysis and discussion of possible confounding factors. All individual points raised are addressed below in a point for point fashion.

Reviewer #1:

The paper is interesting and sheds light on a class of molecules underexplored in context of metabolic diseases and microbiota. However, in my view there are several controls that are missing, a better description of the glucose phenotype is needed, and the evidence concerning the mechanism is too descriptive.

Author response: We thank the reviewer for their interest in our paper and their extremely helpful critique of our study. We have carried out extensive new experimentation aimed at addressing the specific critiques and recommendations. These are outlined in a point for point fashion below.

Main points:

- What are the organs responsible for the improved glucose phenotype in the *Il36rn* KO mice? Hyperinsulinemic-euglycemic clamp, or at least a 2-deoxyglucose tracer studies coupled with pAKT measurements could help elucidating this. The insulin sensitivity should be also investigated in the mice on chow diet (shown in Figure 2).

Author response: As suggested, we have included data demonstrating improved insulin sensitivity in aged mice, now shown in Figure 2f & g. We agree with the reviewer that identifying the organs responsible for the improved glucose phenotype observed would add to the study. Although, we were unable to carry out clamp studies and 2-deoxyglucose tracer studies as suggested, due to technical constraints and a lack of ethical and regulatory approval for such studies, we did examine relative insulin sensitivity in wt and *il36rn*^{-/-} mice subjected to HFD by examining levels of p-Akt in the livers of mice 10 mins after insulin challenge. These new data, included as Figure 3i, demonstrate that insulin sensitivity is indeed increased in the livers of *Il36rn*^{-/-} mice under these conditions.

- To support the claims that the elevated IL36 levels and signalling are important in regulating the metabolic homeostasis, the IL36 alpha, beta and gamma need to be supplemented alone, or in combination in the WT and in the KO mice. This is a critical experiment that could also suggest therapeutic relevance of these findings.

Author response: We agree with the reviewer that supplementation with IL-36 cytokines should be investigated to support the manuscript and point towards therapeutic relevance. Indeed, as suggested, we carried out further experimentation to determine whether the administration of recombinant IL-36 cytokines (a combination of IL-36 α , β , and γ (500ng of

each/ per mouse)) in 5 doses, every other day for 8 days, would improve glucose tolerance in wt mice which had been on a HFD for 12 weeks. The dose of recombinant cytokine administered was chosen based upon recent studies in the literature (Scheibe, K. et al., Gastroenterology, 2019). However, we were unable to demonstrate any effect on glucose tolerance in IL-36 ligand treated mice, over that observed for control mice in these experiments (see figure 1 attached with this rebuttal below). While this lack of an effect is disappointing, it may stem from a relatively short half-life of IL-1 family ligands in vivo, such has been reported for IL-18 and IL-1 β (Hosohara, K et al., Clin Diagn Lab Immunol. 2002)(Kudo, S. et al., Cancer Res, 1990). In addition, it is conceivable that administered cytokines would have to be directly delivered to the colonic mucosa in order to alter the tissue microenvironment there, and influence the pathogenesis of metabolic disease in mice.

- Authors dedicate substantial effort in speculating the mechanisms related to the IL36, but this is not sufficiently addressed experimentally. Does supplementing WT mice with IL-9 promote a KO-like phenotype on the abundance of Akkermansia? Or conversely, does CD4+ depletion in the KO mice reduce mucin (and therefore A. muciniphila's) levels?

Author response: We agree with the reviewers assertion that the mechanisms described in our study could have been addressed more fully experimentally. Guided by their suggestions, we have undertaken significant new experimentation to improve this central part of our study. Firstly, in order to further investigate the role of IL-9 in mediating this phenotype we chose to administer an anti-IL-9 neutralising mAb to determine whether this approach would reverse the observed effects on mucin levels/goblet cell numbers. We chose this approach, ahead of administering recombinant IL-9 as suggested, given our lack of success with IL-36 α,β,γ supplementation as described above. Importantly, administration of this Ab did not alter the relative goblet cell hyperplasia observed in *Il36rn*^{-/-} mice suggesting that elevated IL-9 does not contribute significantly to the phenotype observed. In contrast, administration of recombinant IL-36Ra, not only reversed the outgrowth of A. muciniphila, but also reversed the increased numbers of goblet cells observed in *Il36rn*^{-/-} mice. Furthermore, in extension of these observations, we have also now demonstrated that IL-36 supplementation in vitro led to significant increase in *Muc2* gene expression in wt colon explant cultures after 4hrs. We believe that these new data add significantly to the mechanistic analysis component of our study and indicate a direct role for IL-36 in mediating these effects as opposed to indirectly through increased IL-9. These new data are now shown as an additional figure 6 in the revised manuscript. Given the apparent lack of a role for IL-9, this data is now shown in the new supplementary Figure 4. In addition, we have now also demonstrated that *Il36rn*^{-/-} mice display enhanced intestinal barrier function consistent with the observed changes in the colonic tissue environment (now Fig. 5f).

- The Verucomicrobia in the WT mice shown in Fig. 4c is almost not present. This is surprising, and is not consistent with the literature. These inconsistencies might explain the remarkable increase of the Verucomicrobia abundance (> 1600 fold) in the KO mice. These results need to be confirmed in an independent groups of mice (and preferably also at different ages). The interpretation of these results is difficult without specifying the exact age of the mice – and this should be done in all the figures throughout the study.

Authors Response: We thank the reviewer for their suggestion to specify the exact age of mice included in the studies and we have now addressed this at relevant points throughout the manuscript. We have also analysed the relative abundance of the Verrucomicrobia strain, *A.muciniphila* by qPCR across multiple cohorts of mice and a range of ages as suggested (see Figure 2 attached to this rebuttal). We agree that we observe a low abundance of Verrucomicrobia in the faeces of WT mice in our facility. Our multiple analyses have consistently shown that the levels of Verrucomicrobia, analysed by 16S RNA sequencing, and *A.muciniphila* analysed by qPCR, are found in relatively low abundance in wild type mice in our facility. A similar recent study demonstrated that *Il33*^{-/-} mice have an approx. 2,500 fold increase in relative abundance in *A.muciniphila* in the intestinal microbiome (Malik, A, et al. JCI 2016). While this study did not specify the exact levels of abundance of Verrucomicrobia in WT mice, their data highlights that IL-1 family members can influence the relative abundance this bacterial phylum to a similarly high degree. Although there are reports of higher abundance in the literature, several further analyses are also consistent with our observations of abundance lower than 0.5% (Seregin, S. et al. Cell Rep., 2017). (Xiaofei X, Micro. Research, 2015)(Langille MG, Microbiome, 2014. Moreover, the relative abundance of specific phyla detected in the intestinal microbiome of mice is recognized as being influenced by the specific facility in which they are housed (Ericsson, A.C. et al., Sci Rep. 2018, Rausch, P. et al., Int J Med Microbiol. 2016). While this is a possible confounding factor in our study, as well as in all microbiome research, it is unfortunately something which could not be addressed within the scope of this study.

-The colon mucin layer should be quantified in the WT vs. the KO mice.

Authors Response: We thank the reviewer for this suggestion, which is directly relevant to our microbiome data. We have now carried out this analysis and included new data as Figure. 5 b&d.

Additional points:

- Authors need to confirm that the *Il36rn* KO mice indeed show increased IL36 downstream signalling.

Authors Response: As suggested we have confirmed that signalling pathways downstream of IL-36 are indeed increased in the colons of *il36rn*^{-/-} mice. These data are now included as Figure 4b.

-The study suggests that there is no difference of the macrophage subsets per gram tissue. Notably, despite the lack of calculated significance, in some cases there more than a double fold difference. Based on what power calculation were these animal numbers chosen? Each value should be presented as a data point, and I suggest increasing the animal number (coupled to power calculations) to allow proper interpretation of these results.

Authors Response: We thank the reviewer for this suggestion. The animal numbers chosen for these experiments (n=5) were based upon our experience in carrying out the same types of analysis of macrophage subsets in adipose tissue as previously reported in Hams, E. et al. FASEB Journal, 2016. We have now presented data as individual data points, which as suggested, allows a clearer interpretation of the results.

- In relation to the above comments - there is a large discrepancy between the levels of *A. muciniphila* detected in Fig 4D,E with the ones in figure 6D. Is there an error in this figure?

Authors Response: We apologise for the error made in the scale on the Y axis in Figure 4E as presented in the original manuscript. We thank the reviewer for bringing this to our attention. We have now reanalysed this qPCR data alongside additional controls and corrected the scale in new figure 4f. We have also included new data to original Figure 6d to demonstrate the temporal loss of *A. muciniphila* outgrowth after cohousing using correct scales. See new Figure 7d.

Reviewer #2:

This study identified that IL-36 may protect against metabolic dysfunction in both human and mice. Mechanistically, the authors suggested that the protective effects are dependent on the metabolically protective bacteria *A. muciniphila*. Overall, this manuscript reported an interesting and novel host-microbiota interaction that promotes metabolic health. However, there are some moderate concerns that the authors could address.

1. In Fig. 4E, the *A. muciniphila* level at the various time points in WT and IL-36 $\text{rn}^{-/-}$ mice without IL-36Ra treatment should be provided as controls. This is particularly important as rIL-36Ra injections seemed to bring up the *A. muciniphila* level in WT mice (the opposite of what you would expect from their results).

Authors Response: We thank the reviewer for this important comment. As suggested, we have now included control treatment conditions, and increased animal numbers, to new figure 4f, which more clearly illustrates that rIL-36Ra administration led to a reduction in *A. muciniphila* outgrowth in il36 $\text{rn}^{-/-}$ mice faeces.

2. Fig. 6 co-housing result provided a strong positive association of *A. muciniphila* level and the metabolically protective phenotype. To prove a causal effect of *A. muciniphila*, the authors could gavage mice with *A. muciniphila* and examine whether this approach can produce protective phenotype. This would be the most convincing data for their claim.

Authors Response: We thank the reviewer for this insightful comment. We agree that gavaging mice with *A. muciniphila* would be a strong supporting approach for our study. However, this approach has already proven successful in several recent elegant studies from different groups, confirming a protective role in obesity driven metabolic dysfunction. We believe these studies, which we have cited in support of our investigations (refs 11,12), add significant support to our claims (Everard et al., PNAS 2013, Plovier et al., Nat. Med. 2017, Shin et al., Gut 2014, Li et al. Circulation 2016)

3. IL-36 is classically associated with inflammatory responses. The authors should examine the potential tissue inflammation especially in HFD group to make sure this does not give confounding effect to their findings.

Authors Response: We agree with the reviewers correct assertion that IL-36 is classically associated with inflammatory responses and that this is a potential confounding factor in our study. We now acknowledge this in our discussion section. As suggested, we have also taken steps to examine this issue in HFD exposed mice, through examining levels of serum TNF and IL-6 between both control and *Il36rn*^{-/-} mice. These new data, shown in supplementary figure 3, demonstrate that there is no significant difference in levels of these proinflammatory mediators detected between wt and *Il36rn*^{-/-} mice.

Reviewers Figure 1. Administration of IL-36 α + β + γ cocktail to wt mice after HFD does not alter glucose tolerance. C57BL/6 mice were placed on HFD for 12 weeks. In week 13 mice were either administered 5 doses of a combination of IL-36 α , IL-36 β , and IL-36 γ (500ng of each cytokine) (n=5) or PBS control (n=5) by i.p. injection every other day for 8 days. On day 10 both groups of mice were subjected to glucose tolerance test. Data shown as glucose over time after i.p. glucose injection and AUC. Statistical analysis using two-tailed unpaired student's t-test for AUC, or RM two-way ANOVA.

Reviewers Figure 2. A. muciniphila abundance in different cohorts of wt and *Il36rn*^{-/-} mice across different ages. Data represent means \pm SEM. Statistical analysis using two-tailed unpaired students t-test.

Reviewers' comments:

Reviewer #1 (Remarks to the Author):

Authors responded to most of my concerns, however the glucose phenotype is still insufficiently addressed. In Fig 3i, there seem to be an increase in the loading controls. These WBs need to be quantified and values should be shown as relative pAKT/total AKT. In addition, from the same animals, authors should investigate pAKT and pIR levels in all the insulin sensitive tissues, including the sc and visceral adipose tissues, and the muscle. These quantifications need to be presented in the main figures.

Reviewer #2 (Remarks to the Author):

After a thorough review, I found that the authors have adequately addressed all my comments. They have new data showing proper controls in Fig 4f as well as data in supp. fig 3 to show there is no significant difference in level of proinflammatory cytokines such as IL-1, IL-6, TNF-a etc between WT and I136rn^{-/-} mice. I believe this paper should now be published in Nature Communications.

Hsin-Jung Joyce Wu

Response to Reviewers' comments:

We thank the reviewers for their very positive comments about our revised manuscript and suggestions for further improvements. Based upon these suggestions, we have undertaken further additional experimentation and analysis as outlined below. Having been guided by the thorough reviews of our data, we now believe that our manuscript is substantially improved since first submission.

Reviewer #1:

Authors responded to most of my concerns, however the glucose phenotype is still insufficiently addressed. In Fig 3i, there seem to be an increase in the loading controls. These WBs need to be quantified and values should be shown as relative pAKT/total AKT. In addition, from the same animals, authors should investigate pAKT and pIR levels in all the insulin sensitive tissues, including the sc and visceral adipose tissues, and the muscle. These quantifications need to be presented in the main figures.

Author response:

We thank the reviewer for their suggestion to incorporate a more comprehensive analysis of the glucose phenotype in our manuscript. As suggested, we have now investigated pAKT and pIR levels in muscle and adipose tissue, from the same animals, and quantified WB data through densitometry to analyse relative levels of activity of AKT and IR in all three tissue types. These quantifications and data are now included in a revised figure 3 (and source data file) and demonstrate that enhanced insulin sensitivity is only evident in liver tissues of *Il36rn^{-/-}* mice upon insulin challenge.

REVIEWERS' COMMENTS:

Reviewer #1 (Remarks to the Author):

Authors addressed my remaining comments.

Response to Reviewers' comments:

We thank the reviewers for their confirmation that we have addressed their remaining comments.